# Application of Quantitative Computed Tomographic Perfusion in the Prognostic Assessment of Patients with Aneurysmal Subarachnoid Hemorrhage Coexistent Intracranial Atherosclerotic Stenosis

**DOI:** 10.3390/brainsci13040625

**Published:** 2023-04-06

**Authors:** Jun Yang, Heze Han, Yu Chen, Fa Lin, Runting Li, JunLin Lu, Ruinan Li, Zhipeng Li, Guangzhi Shi, Shuo Wang, Yuanli Zhao, Xiaolin Chen, Jizong Zhao

**Affiliations:** 1Department of Neurosurgery, Beijing Tiantan Hospital, Capital Medical University, Beijing 100070, Chinachenxiaolin@bjtth.org (X.C.); 2Department of Neurosurgery, West China Hospital, Sichuan University, Chengdu 610000, China; 3Department of Critical Care Medicine, Beijing Tiantan Hospital, Capital Medical University, Beijing 100070, China; 4China National Clinical Research Center for Neurological Diseases, Beijing 100070, China; 5Center of Stroke, Beijing Institute for Brain Disorders, Beijing 100070, China; 6Beijing Key Laboratory of Translational Medicine for Cerebrovascular Disease, Beijing 100070, China

**Keywords:** aneurysmal subarachnoid hemorrhage, intracranial atherosclerotic stenosis, delayed cerebral ischemia, computed tomography perfusion

## Abstract

The comorbidity of aneurysmal subarachnoid hemorrhage (aSAH) with intracranial atherosclerotic stenosis (ICAS) has been suggested to increase the risk of postoperative ischemic stroke. Logistic regression models were established to explore the association between computed tomography perfusion (CTP) parameters and 3-month neurological outcomes and delayed cerebral ischemia (DCI). Prognostic-related perfusion parameters were added to the existing prognostic prediction models to evaluate model performance improvement. Tmax > 4.0 s volume > 0 mL was significantly associated with 3-month unfavorable neurological outcomes after adjusting for potential confounders (OR 3.90, 95% CI 1.11–13.73), whereas the stenosis degree of ICAS was not. Although the cross-validated area under the curve (AUC) was similar after the addition of the Tmax > 4.0 s volume > 0 mL (SAHIT: *p* = 0.591; TAPS: *p* = 0.379), the continuous net reclassification index (cNRI) and integrated discrimination index (IDI) showed that the perfusion parameters significantly improved the performance of the two models (*p* < 0.001 for all comparisons). Patients with coexistent aSAH and ICAS, Tmax > 4.0 s volume > 0 mL is an independent factor of 3-month neurological outcomes. A quantitative assessment of cerebral perfusion may help accurately screen patients with poor outcomes due to the coexistence of aSAH and ICAS.

## 1. Introduction

Aneurysmal subarachnoid hemorrhage (aSAH) is associated with a mortality of more than 30%. Only about 30% of patients with aSAH recover sufficiently to return to independent living [1]. The coexistence of intracranial aneurysms and intracranial atherosclerotic stenosis (ICAS) is an unusual phenomenon, but the presence of comorbidities complicates the management of the disease, especially in patients with aSAH [2,3].

In clinical scenarios, the coexistence of aSAH and internal carotid artery stenosis pose great challenges to clinical treatment decision-making, especially when two diseases occur in the same vessel. On the one hand, the neurosurgeon should consider the risk of hemorrhagic stroke caused by the re-rupture of the aneurysm responsible for aSAH. On the other hand, the risk of ischemic stroke is caused by aggravated cerebrovascular stenosis [4]. Therefore, the coexistence of aSAH and ICAS is extremely difficult in clinical management. The key to obtaining a good prognosis for these patients is how to deal with the culprit aneurysm while avoiding the occurrence of ischemic stroke.

Delayed cerebral ischemia (DCI) occurs in approximately 30% of aSAH patients after aneurysm rupture and is the main contributing factor to mortality and patients who had long-term handicaps after aSAH [5]. Therefore, aSAH patients complicated with ICAS may have a higher risk of postoperative DCI due to the additive effect [6]. However, how to accurately identify high-risk patients with DCI among patients with aSAH complicated with ICAS remains an urgent problem to be solved. Previous studies have found that early cerebral hypoperfusion after aneurysm rupture is significantly associated with DCI after aSAH [7,8,9]. Therefore, the quantitative assessment of cerebral perfusion may be helpful for the early identification of perioperative DCI in aSAH. Quantitative analysis of CT perfusion (CTP) is an effective technique for the early identification of high-risk patients with DCI [10].

Given that the comorbidity of aSAH and ICAS is not uncommon, its prognosis must be clarified, and high-risk patients with poor outcomes must be accurately identified. Therefore, this study aimed to clarify the perioperative complications and long-term outcomes of the aSAH-ICAS comorbidity, and to further investigate the value of CTP parameters in a prognostic assessment.

## 2. Methods

### 2.1. Study Design, Setting, and Data Sources

This case–control study used a multicenter prospective registry data from the Long-term Prognosis of Emergency Aneurysmal Subarachnoid Hemorrhage (LongTEAM study, ClinicalTrials.gov, NCT 04785976) to explore the available prognostic-related cerebral perfusion parameters for patients with coexistent aSAH and ICAS (Appendix B). This study was conducted following the STROBE reporting guidelines.

The LongTEAM study is a multicenter prospective registry to improve the diagnosis, treatment effect, and efficiency in aSAH while opening up new avenues for interdisciplinary clinical practice and scientific research exploration. Previous studies have demonstrated the database’s validity and quality for research [11]. This study was carried out according to the guideline of the 1964 Helsinki Declaration and was approved by the institutional ethics committee (IRB approval number: KY 2021-008-01). Written informed consent was obtained at admission before entering the study.

### 2.2. Study Population

All aSAH patients included in the LongTEAM study between 1 January 2019 and 1 September 2022 were reviewed to identify patients who underwent microsurgical clipping or endovascular coiling and co-existing ICAS. The presence of ICAS was confirmed by imaging (CT angiography (CTA) or digital subtraction angiography (DSA)). The details are as follows: the inclusion criteria: (1) patients who were diagnosed with aSAH by CT and DSA or CTA; (2) patients who were diagnosed with ICAS by CTA or DSA and underwent CTP within 24 h of admission; (3) adult patients > 18 years of age; (4) the aneurysms were treated by clipping or coiling; (5) SAH from diagnosis to surgical treatment within seven days. Patients with congenital cerebral vascular disease (e.g., arteriovenous malformations and moyamoya disease) and coexistent intracranial lesions were simultaneously treated (e.g., resection of meningioma or pituitary adenoma) were excluded from the present study.

### 2.3. Baseline Characteristics

Demographic factors (age at diagnosis, sex), clinical presentation (Hunt–Hess (H-H) grade, World Federation of Neurosurgical Societies (WFNS) grade, etc.), morphological features (aneurysm characteristics, stenosis degree, image grading, etc.), treatment strategy, perioperative complication, and neurological status were collected in this study. 

The clinical severity of aSAH was assessed using HH and WFNS grades at the initial presentation. Modified Fisher Scale (mFS), Graeb score, and Subarachnoid hemorrhage Early Brain Edema Score (SEBES) were used for imaging evaluation based on a CT scan at admission [12]. The degree of intracranial stenosis measured by CTA was calculated using the published method from the WASID study, and the severity grading was referenced to the Tiziano (mild: <70%; moderate: 70–99%; severe: >99%) [13,14]. According to the Trial NASCET which began in 1987, less than 70 percent of the luminal diameter stenosis is defined as moderate stenosis. Those with severe stenosis are defined as having stenosis of 70 to 99 percent [15]. We chose Tiziano for more stringent indications for inclusion.

### 2.4. CTP Parameter and RAPID

CTP was performed using a 256-slice CT scanner (Brilliance iCT, Philips Healthcare, Cleveland, OH, USA) with the injection of 50 mL of iodinated contrast material (iohexol [Omnipaque], 350 mg of iodine per milliliter) into an antecubital vein at a rate of 6 mL/s, followed by 20 mL of saline flush at the same injection rate. The acquisition parameters were set as follows: Jog mode, 80 kVp, 300 mAs, 256*0.625 mm collimation, 1 s rotation time, and 5 mm slice thickness at the 0 mm interval. A total of 480 slices were obtained with a 160 mm scan length and a scan time of 60 s. CTA was reconstructed with a 0.90 mm thickness and 0.50 mm increment based on original CTP voxel information.

Perfusion parameters with different thresholds were automatically estimated with the RAPID software (iSchema View Inc., Menlo Park, CA, USA), which is nowadays largely adopted for detecting the infarct core and the ischemic penumbra among large vessel occlusion patients [16]. The software can create brain perfusion maps indicating several parameters such as cerebral blood volume (CBV), cerebral blood flow (CBF), mean transit time (MTT), and time-to-maximum of the tissue residue function (Tmax).

### 2.5. Outcome Assessment

The primary outcome was the modified Rankin scale (mRS) at 90 days after discharge (the neurosurgeon followed up with patients via telephone or an outpatient appointment). The mRS was dichotomized into favorable clinical outcomes (mRS ≤ 2) and unfavorable outcomes (mRS > 2). The secondary outcomes were the occurrence of DCI in hospitalization—defined as clinical deterioration—a new focal neurologic deficit, or a new infarction on CT that is not attributable to other causes [5].

### 2.6. Statistical Analysis

Categorical variables were presented as percentages and continuous variables as mean with standard deviation (SD) or median with interquartile range (IQR). In the univariate analysis of prognostic factors based on mRS and DCI, the two-tailed *t*-test or Mann–Whitney test was used for continuous variables. The χ^2^ test or Fisher’s exact test was utilized for the categorical variables as appropriate.

To investigate the specific CTP parameters associated with the primary outcomes (3-month mRS), a multivariable logistic regression model was performed on these parameters separately, adjusting for both the statistically and the clinically significant factors. The results were expressed as odds ratio (OR) with 95% CI. The parameters remaining significant in all the adjustments were selected for further analysis. Further, prespecified subgroup analyses of prognostic-related perfusion parameters were performed, categorizing by treatment modality (surgical clipping or endovascular treatment), degree of stenosis (mild, moderate, or severe), sex (female or male), and age (≥60 years or <60 years). Tests of interaction were also performed in the categories mentioned above. Based on two previous prognostic models (Subarachnoid Hemorrhage International Trialists’ model (SAHIT) and Tiantan Aneurysmal Subarachnoid Hemorrhage Prognostic Scoring System (TAPS)) [17,18], we further supplemented these two models by adding prognostic-related perfusion parameters. These models were trained using 10-fold cross-validation with the procedure iterated 100 times to reduce the risk of overfitting due to the choice of folds. To address the issue of imbalanced outcome distribution, the random under-sampling procedure was implemented. This technique involved randomly selecting a subset of samples from the majority class to match the size of the minority class, which can lessen the potential bias of models toward the majority class. The area under the curves (AUCs) was compared using the DeLong test. Continuous net reclassification index (cNRI) and integrated discrimination index (IDI) were calculated to see whether the CTP parameters added more predictive value to the two models in aSAH patients co-existing with ICAS.

All statistical analyses were performed using R version 4.0.3 (R Foundation for Statistical Computing). *p* values were 2-sided, and *p* < 0.05 was considered statistically significant.

## 3. Results

### 3.1. Study Population and Baseline Characteristics

Of the 931 aSAH patients treated by microsurgical clipping or endovascular coiling for aSAH in the LongTEAM study between 1 January 2019 and 1 September 2022, 233 aSAH patients admitted within seven days of onset had co-existing ICAS after excluding 41 cases who did not have CTP examination within 24 h before surgery, yielding an overall cohort of 192 aSAH patients with co-existing ICAS for this study. Figure 1 shows the enrollment process. There was no significant differences between the CTP-group and the no CTP-group (Appendix A).

The mean age was 59.34 ± 10.77 years, and the gender distribution was balanced (female: 57.3%) (Table 1). A total of 35 (18.2%) were classified as H-H grade 4–5, 59 (30.7%) were WFNS grade 4–5, 130 (67.7%) were mFS grade 3–4, 21 (10.9%) were Graeb score 5–12, and 79 (41.1%) were SEBES score 3–4. Two hundred and twenty-nine aneurysms were found in 192 patients, with an average of 1.19 aneurysms. All 192 culprit aneurysms were saccular in morphology, and the internal carotid artery is the most common parent artery (82/192, 42.7%), following the middle cerebral artery (38/192, 19.8%). The average aneurysm dome size was 6.2 ± 3.9 mm. For the ICAS, about half of them (88, 45.8%) were parent artery stenosis. When considering the stenosis degree, 139 (72.4%) were mild stenosis (<70%), 33 (17.2%) were moderate stenosis (70–99%), and 20 (10.4%) were severe stenosis (>99%). Finally, 102 (53.1%) were treated with microsurgical clipping and 90 (46.9%) with endovascular coiling.

### 3.2. Association between Perfusion Parameter and 3-Month mRS

At the 3-month follow-up after enrollment, 42 (21.9%) obtained unfavorable neurological outcomes (mRS > 2), 67 (34.9%) developed DCI, and 12 (6.3%) died from complications. In the univariate analysis, hypertension, early loss of consciousness, H-H grade 4–5, WFNS grade 4–5, all CTP parameters, treatment strategy, and most perioperative complications were associated with 3-month mRS. Interestingly, the stenosis degree of ICAS was not correlated with 3-month mRS (mRS > 2) (Table 2). Therefore, in further univariate and multivariate logistic regression analysis, we mainly explored the prognostic correlation of perfusion parameters obtained based on CTP (adjusted for stenosis degree as one of the covariables in the multivariate logistic regression model).

The univariable logistic regression analysis showed that all the CTP parameters were significantly associated with 3-month mRS (mRS > 2) (Table 3). After adjusting for both statistically and clinically significant factors in the multivariable logistic regression model, only Tmax > 4.0 s volume > 0 mL (OR 3.90, 95% CI 1.11–13.73) remained significantly associated with 3-month mRS (mRS > 2). Further, in investigating whether CTP parameters were related to the occurrence of DCI, no perfusion parameters showed sufficient correlation after adjusting for potential statistical and clinical correlation parameters. Given the possible slightly broad defects of Tmax > 4.0 s volume > 0 mL, the upper quartile of Tmax > 4.0 s volume (Tmax > 4.0 s volume > 25 mL) was selected as the covariable for further adjustment. Still, it did not correlate well with the 3-month neurological outcomes. Finally, the Tmax > 4.0 s volume > 0 mL was selected to predict 3-month neurological outcomes for further analysis.

### 3.3. Prespecified Subgroup Analysis and Additive Value Assessment

In the prespecified subgroup analysis, the positive association was not modified by treatment modality, sex, and age; there was no significant interaction between the perfusion parameter and these subgroups (Figure 2). The effect sizes of the volume in the moderate and severe intracranial arterial stenosis subgroups could not be calculated as all the patients with unfavorable outcomes were in the Tmax > 4.0 s volume > 0 mL group.

The cross-validated AUC did not differ significantly after adding the Tmax > 4.0 s volume > 0 mL (SAHIT: *p* = 0.591; TAPS: *p* = 0.379). The evaluation metrics (sensitivity, specificity, accuracy, positive predictive value, and negative predictive value) derived from the confusion matrix of the four models were provided in Appendix A. After the random under-sampling procedure, the overall sensitivity and specificity of models were balanced, and still, no significant addictive value was observed in these metrics (Appendix A). However, the cNRI and IDI showed that the perfusion predictor significantly improved the performance of the two models (*p* < 0.001 for all comparisons, Table 4).

## 4. Discussion

The prognostic value of perfusion parameters in aSAH co-existing with ICAS remains unclear. This study found that the stenosis degree of ICAS was not correlated with the neurological outcomes at 3 months after the onset of aSAH, while the CTP-based Tmax > 4.0 s volume > 0 mL was a significant factor. On this basis, Tmax > 4.0 s volume > 0 mL was further confirmed to improve the efficacy of previous prognostic models with aSAH significantly. This study demonstrated the value of CTP-based perfusion parameters rather than stenosis degree in predicting prognosis in patients with coexistent aSAH and ICAS.

Intracranial atherosclerosis is an important cause of ischemic stroke and stroke recurrence, compared with the other stroke subtypes [19]. Most previous studies focused on carotid artery stenosis in patients coexistent with unruptured intracranial aneurysms and whether performing carotid endarterectomy or carotid artery stenting affects the safety of the aneurysm. At the same time, there has been a lack of consensus regarding indications on the safety and treatment strategies for aSAH coexistent with ICAS [4,13,20].

In aSAH, cerebral ischemia is a major cause of morbidity and mortality [21]. Generally, the rapid extravasation of blood after an aneurysm rupture will lead to a significant increase in intracranial pressure induced by acute hypoperfusion, leading to the continuous damage of cerebral perfusion pressure and a decrease in cerebral blood flow, eventually leading to cerebral ischemia and neuronal necrosis [22,23,24]. Therefore, the question before neurosurgeons is whether patients with ruptured aneurysms have a higher probability of postoperative cerebral ischemia when ICAS is found in preoperative CTA.

In this study, we adopted the vascular stenosis evaluation criteria based on Tiziano to classify the stenosis degree of ICAS into three categories (mild: <70%; moderate: 70–99%; severe: >99%) [13,14]. Normally, moderate or even severe ICAS would lead to intracranial hypoperfusion, intraoperative manipulation, and blood pressure fluctuations during anesthesia, leading to insufficient cerebral perfusion and increasing the risk for perioperative strokes [25]. However, surprisingly, we did not find a correlation between the stenosis degree of ICAS and the 3-month neurological outcomes of aSAH. Those vessels with moderate-to-severe stenosis have collateral circulation as a substitute, even if the arteries are occluded [26]. Cross-collateral compensation of flow through the circle of Willis and additional compensatory intracerebral vasodilation may mitigate the effects of stenosis-induced flow restriction [6].

Nowadays, CTP is widely used to assess cerebral perfusion, and previous studies consider early stage CTP in aSAH as a promising tool for predicting perioperative DCI [27]. Tmax volume hypoperfusion is largely adopted for detecting the infarct core and ischemic penumbra among large vessel occlusion patients [28]. Several studies identified Tmax > 10 s as an important threshold to predict poor prognosis in acute ischemic stroke [29]. However, in aSAH with hemorrhagic stroke, this study found that Tmax was only affected to a small extent after the onset, and patients with Tmax > 6 s, 8 s, and 10 s equal to 0 mL accounted for the majority (68.2%, 72.4%, 76.0%), while 66.7% of patients had abnormal Tmax > 4 s. In this study, the intracranial perfusion of aSAH was quantitatively analyzed using perfusion parameters automatically estimated by the RAPID software. Finally, Tmax > 4.0 s volume > 0 mL was found to be an independent factor associated with 3-month unfavorable neurological outcomes (mRS > 2).

SAHIT and the TAPS score system have been published for the prognosis prediction of patients with aSAH. In [17,18], we compared the models before and after adding the perfusion parameter (Tmax > 4.0 s volume > 0 mL) to investigate the performance of the new models. The cNRI and IDI showed the superiority of the new model in predicting the 3-month mRS. This means that in patients with aSAH co-existing with ICAS, a comprehensive assessment by multimodal imaging (e.g., CTP-based perfusion assessment) can help to more accurately identify high-risk patients with unfavorable outcomes, thus encouraging clinicians to take early preventive measures to improve neurological outcomes.

Near-infrared spectroscopy (NIRS) is a noninvasive technique that can provide a surrogate marker of changes in CBF and used to assess cerebral autoregulation in adults after subarachnoid hemorrhage [30]. It seems likely that NIRS and CTP will become a clinical tool in the foreseeable future, which will enable diagnosis and assessments in aneurysmal subarachnoid hemorrhages.

### Limitations

Our study had several limitations. First, the diagnosis of ICAS is easily confused with the high incidence of cerebral vasospasm in the perioperative period of aSAH, leading to the misdiagnosis of ICAS. To avoid this bias, the CTA examination results of the patients three months after the onset were used for verification to prevent misdiagnosis. Second, although we have made many improvements in the measurement method of stenosis degree based on the WASID method, the measurement results are still inaccurate [31]. We repeat the measurements on the CTA from as many angles as possible. Third, for patients with aSAH co-existing with multiple ICAS, especially those with bilateral ICAS, clinical researchers are sometimes unable to judge which side of the ICAS is the real responsible lesion that may affect the occurrence of DCI. The resulting misjudgment may be the potential reason that the degree of ICAS stenosis has nothing to do with the prognosis in this study. To reduce the resulting bias, we usually select the most severe stenosis vessel segment as the culprit lesion when multiple segmental stenoses are encountered. Fourth, Tmax > 4.0 s volume > 0 mL is a broad range. In future studies, the sample size should be increased to find a more accurate Tmax-based volume threshold for evaluating perfusion imbalance brain tissue, which will better promote the precise early identification of high-risk groups with poor prognosis in patients with aSAH co-existing with ICAS. Fifth, the results should be interpreted with caution as the conclusions were drawn after adjusting for numerous potential confounders, while the outcome events of this study did not meet the event per variable principle. The reliability and robustness of the results require further validation with a larger sample size.

## 5. Conclusions

In patients with coexistent aSAH and ICAS, Tmax > 4.0 s volume > 0 mL is an independent factor of 3-month neurological outcomes. The quantitative assessment of cerebral perfusion may help accurately screen patients with poor outcomes due to the coexistence of aSAH and ICAS.

## Figures and Tables

**Figure 1 brainsci-13-00625-f001:**
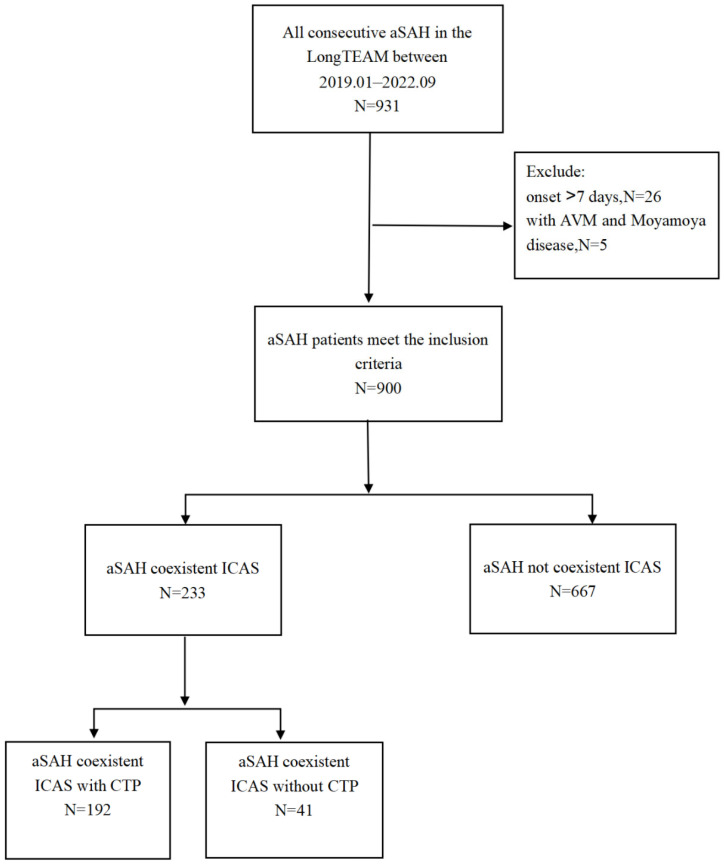
Flow diagram of the enrolled patients.

**Figure 2 brainsci-13-00625-f002:**
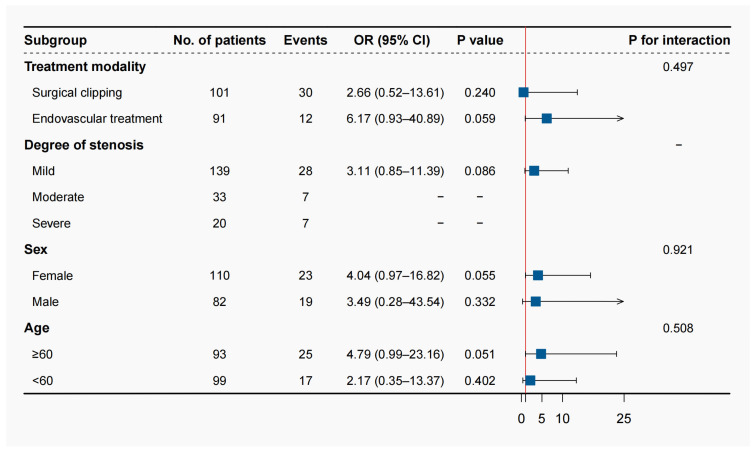
Prespecified subgroup analysis.

**Table 1 brainsci-13-00625-t001:** Baseline characteristics of patients with coexistent aSAH and ICAS.

Patients	Value
Total no.	192
Average age in years (SD)	59.3 (10.8)
Sex, Female/Male	1:1.34
Neurological score	
H-H grade 4–5	35 (18.2%)
WFNS grade 4–5	59 (30.7%)
mFS grade 3–4	130 (67.7%)
Graeb score 5–12	21 (10.9%)
SEBES score 3–4	79 (41.1%)
Degree of stenosis	
>99%	20 (10.4%)
70–99%	33 (17.2%)
<70%	139 (72.4%)
Average dome size in mm (SD)	6.2 (3.9)
Location	
ICA	82 (42.7%)
ACoA	36 (18.8%)
MCA	38 (19.8%)
ACA	11 (5.7%)
PICA	4 (2.1%)
Treatment	
Clipping	102 (53.1%)
Coiling	90 (46.9%)
Periprocedural complication	
DCI	67 (34.9%)
Mortality	12 (6.3%)

Abbreviations: ACA, anterior cerebral artery; AcoA, anterior communicating artery; aSAH, aneurysmal Subarachnoid Hemorrhage; DCI, delayed cerebral ischemia; H-H, Hunt–Hess; ICA, internal carotid artery; ICAS, intracranial arterial stenosis; MCA, middle cerebral artery; mFS, modified Fisher scale; PICA, posterior inferior cerebellar artery; SD, standard deviation; SEBES, Subarachnoid hemorrhage Early Brain Edema Score; WFNS, World Federation of Neurosurgical Societies.

**Table 2 brainsci-13-00625-t002:** Univariate analysis of baseline characteristics of mRS and DCI.

Characteristics	mRS Score ≤ 2	mRS Score > 2	*p* Value	DCI	Non-DCI	*p* Value
Sample size	150 (78.1%)	42 (21.9%)		67 (34.9)	125 (65.1)	
Age, years, mean ± SD	58.29 ± 10.33	63.12 ± 11.57	0.010	58.27 ± 10.85	59.92 ± 10.73	0.313
Female sex	87 (58.0)	23 (54.8)	0.843	42 (62.7)	68 (54.5)	0.269
Prior medical history						
Smoking	18 (12.0)	6 (14.3)	0.922	13 (19.4)	16 (12.8)	0.223
Drinking	8 (5.3)	5 (11.9)	0.196	7 (10.4)	8 (6.4)	0.319
Hypertension	82 (54.7)	31 (73.8)	0.040	45 (67.2)	68 (54.4)	0.087
Diabetes mellitus	13 (8.7)	3 (7.1)	1.000	4 (6.0)	12 (9.6)	0.386
Prior infarction	14 (9.3)	1 (2.4)	0.913	3 (4.5)	12 (9.6)	0.207
Aneurysm characteristics						
Posterior circulation	20 (13.3)	7 (16.7)	0.766	9 (13.4)	18 (14.4)	0.854
Early seizures	5 (3.3)	2 (4.8)	1.000	3 (4.5)	4 (3.2)	0.653
Early loss of consciousness	36 (24.0)	25 (59.5)	<0.001	28 (41.8)	33 (26.4)	0.029
Acute hydrocephalus	67 (44.7)	16 (38.1)	0.559	30 (44.8)	53 (42.4)	0.751
Parent artery stenosis	67 (44.7)	21 (50.0)	0.661	35 (52.2)	53 (42.4)	0.192
Neurological score						
H-H grade 4–5	15 (10.0)	20 (47.6)	<0.001	15 (22.4)	20 (16.0)	0.274
WFNS grade 4–5	32 (21.3)	27 (64.3)	<0.001	27 (40.3)	32 (25.6)	0.035
mFS grade 3–4	96 (64.0)	34 (81.0)	0.059	50 (74.6)	80 (64.0)	0.133
Graeb score 5–12	13 (8.7)	8 (19.0)	0.104	6 (9.0)	15 (12.0)	0.519
SEBES score 3–4	60 (40.0)	19 (45.2)	0.665	29 (43.3)	50 (40.0)	0.659
CTP parameters, median (IQR)						
Tmax > 8.0 s volume	0.0 (0.0–0.0)	0.0 (0.0–14.0)	0.016	0.0 (0.0–6.0)	0.0 (0.0–3.5)	<0.001
Tmax > 6.0 s volume	0.0 (0.0–3.0)	4.0 (0.0–43.0)	<0.001	0.0 (0.0–16.0)	0.0 (0.0–6.5)	<0.001
Tmax > 4.0 s volume	12.0 (0.0–73.3)	123.0 (27.5–309.0)	<0.001	35 (0.0–151.0)	17 (0.0–121.5)	<0.001
rCBF < 38% volume	0.0 (0.0–0.0)	0.0 (0.0–20.8)	<0.001	0.0 (0.0–0.0)	0.0 (0.0–0.0)	<0.001
Treatment modality			0.010			0.002
Surgical clipping	71 (47.3)	30 (71.4)		46 (68.7)	56 (44.8)	
Endovascular coiling	79 (52.7)	12 (28.6)		21 (31.3)	69 (55.2)	
Degree of stenosis			0.322			0.872
Mild (<70%)	111 (74.0)	28 (66.7)		48 (71.6)	91 (72.8)	
Moderate (70–99%)	26 (17.3)	7 (16.7)		11 (16.4)	22 (17.6)	
Severe (>99%)	13 (8.7)	7 (16.7)		8 (11.9)	12 (9.6)	

Abbreviations: CTP, computed tomography perfusion; DCI, delayed cerebral ischemia; H-H, Hunt–Hess; IQR, interquartile range; mFS, modified Fisher Scale; mRS, modified Rankin Scale; rCBF, relative cerebral blood flow; SD, standard deviation; SEBES, Subarachnoid hemorrhage Early Brain Edema Score; Tmax, Time-to-maximum of the tissue residue function; WFNS, World Federation of Neurosurgical Societies.

**Table 3 brainsci-13-00625-t003:** Univariable and multivariable logistic analysis of RAPID-based CTP parameters in predicting 3-month mRS * and perioperative DCI #.

	Tmax > 8.0 s Volume > 0 mL	Tmax > 6.0 s Volume > 0 mL	Tmax > 4.0 s Volume > 0 mL	Tmax > 4.0 s Volume > 25 mL	rCBF < 38% Volume > 0 mL
3-month mRS					
Univariable analysis	2.82 (1.37–5.78)	3.13 (1.54–6.35)	4.67 (1.73–12.56) ^a^	3.88 (1.88–8.02)	3.75 (1.76–8.03)
Multivariable analysis 1 †	0.76 (0.27–2.10)	0.85 (0.31–2.39)	3.46 (1.10–10.89) ^a^	1.59 (0.62–4.07)	1.04 (0.36–3.07)
Multivariable analysis 2 ‡	0.76 (0.25–2.31)	0.87 (0.28–2.74)	3.90 (1.11–13.73) ^a^	1.29 (0.43–3.88)	0.67 (0.19–2.34)
Perioperative DCI					
Univariable analysis	1.33 (0.69–2.55)	1.47 (0.78–2.76)	1.23 (0.65–2.34)	2.21 (1.21–4.06)	1.32 (0.64–2.70)
Multivariable analysis 1 §	0.76 (0.36–1.62)	0.82 (0.39–1.73)	0.83 (0.41–1.67)	1.70 (0.86–3.38)	0.66 (0.28–1.53)
Multivariable analysis 2 ¶	0.67 (0.27–1.66)	0.84 (0.34–2.08)	1.00 (0.43–2.31)	2.07 (0.87–4.90)	0.53 (0.19–1.46)

Abbreviations: CTP, computed tomography perfusion; mRS, modified Rankin scale; rCBF, relative Cerebral Blood Flow; Tmax, Time-to-maximum of the tissue residue function. * The CTP parameters were included in the univariable and multivariable analysis separately to avoid multicollinearity. Results were expressed as odds ratio with 95% confidence interval. † The multivariable analysis 1 adjusted for covariates that were statistically significant in the baseline comparisons (sex, history of smoking, history of alcohol consumption, hypertension, early loss of consciousness, Hunt–Hess grade, WFNS grade, modified Fisher Scale, Graeb score, and treatment modality). ‡ The multivariable analysis 2 adjusted for covariates in analysis 1 and those considered significant in clinical practice (hyperlipidemia, diabetes mellitus, SEBES score, parent artery stenosis, and the degree of stenosis). ^a^ Tmax > 4.0 s volume > 0 mL remained significantly associated with 3-month unfavorable outcomes after different adjustments. # The CTP parameters were included in the univariable and multivariable analysis separately to avoid multicollinearity. Results were expressed as odds ratio with 95% confidence interval. § The multivariable analysis 1 adjusted for covariates that were statistically significant in the baseline comparisons (early loss of consciousness, WFNS grade, and treatment modality). ¶ The multivariable analysis 2 adjusted for covariates in analysis 1 and those considered significant in clinical practice (sex, history of smoking, history of alcohol consumption, hypertension, hyperlipidemia, diabetes mellitus, modified Fisher Scale, Graeb score SEBES score, parent artery stenosis, and the degree of stenosis).

**Table 4 brainsci-13-00625-t004:** Performance of previous models after the addition of the CTP parameter.

Model	AUC (95%CI)	*p* Value	cNRI (95%CI)	*p* Value	IDI (95%CI)	*p* Value
SAHIT	0.821 (0.754–0.888)	Reference	Reference	Reference	Reference	Reference
SAHIT + CTP	0.831 (0.763–0.899)	0.591	0.535 (0.284–0.786)	<0.001	0.041 (0.018–0.064)	<0.001
TAPS	0.769 (0.689–0.848)	Reference	Reference	Reference	Reference	Reference
TAPS + CTP	0.790 (0.709–0.871)	0.379	0.541 (0.277–0.805)	<0.001	0.042 (0.021–0.062)	<0.001

Abbreviations: AUC, area under the curve; CI, confidence interval; cNRI, continuous Net Reclassification Index; CTP, computed tomography perfusion, IDI, integrated discrimination improvement; SAHIT, Subarachnoid Hemorrhage International Trialists; TAPS, Tiantan Aneurysmal Subarachnoid Hemorrhage Prognostic Scoring System.

## Data Availability

All original data are available upon reasonable request to the corresponding authors.

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
