# Peer review of "Application of Quantitative Computed Tomographic Perfusion in the Prognostic Assessment of Patients with Aneurysmal Subarachnoid Hemorrhage Coexistent Intracranial Atherosclerotic Stenosis"

_brainsci, 2023, doi:10.3390/brainsci13040625_

Round 1
Reviewer 1 Report
The authors presented a useful work where they pointed the need for quantitative assessment of cerebral perfusion that may be helpful for early identification of perioperative DCI in aSAH. They explained that quantitative analysis of CT perfusion (CTP) is an effective technique for the early identification of high-risk patients with DCI.
I have only few comments:
1. Introduction part line37-38: Please decide for ICAS abbreviation: "artery" or "atherosclerotic" since you wrote overall in the text "artery".
2. It will be nice to visualize these "Perfusion parameters" (CBV, CBF,MTT,Tmax) or brain perfusion maps from the software as additional Figure.
Author Response
Reviewer #1:
Comment 1:
Introduction part line37-38: Please decide for ICAS abbreviation: "artery" or "atherosclerotic" since you wrote overall in the text "artery".
Response 1:
Thanks to your reminder, we have corrected the error and replaced "artery" with "atherosclerotic".
Comment 2:
It will be nice to visualize these "Perfusion parameters" (CBV, CBF,MTT,Tmax) or brain perfusion maps from the software as additional Figure.
Response 2:
Thanks for your comment. We add a classical case report of a patient.The patient was female, 74 years old, admitted to Tiantan hospital as an emergency with a sudden severe headache for 6 hours. The cranial CT indicated subarachnoid hemorrhage(A), and the CTA indicated left posterior communicating artery aneurysm(B), combined with multiple intracranial atherosclerotic stenosis, especially bilateral intracranial segment of vertebral artery and basilar artery(B and C). The patient was evaluated by RAPID software after preoperative CTP examination(D), and the Tmax>4.0s volume =181mL(E). The patient accepted surgical clipping, and tolerated the procedure well. However, on postoperative day 6, the patient had a sudden loss of consciousness followed by bilateral pupil dilatation. Emergency cranial CT indicated multiple infarction in the brainstem and cerebellum in the region supplied by the basilar artery and posterior cerebral artery(F).Despite various therapeutic measures, and the patient's condition continued to deteriorate and eventually died.

Reviewer 2 Report
Postoperative ischemic stroke is associated with aneurysmal subarachnoid hemorrhage (aSAH) and intracranial arterial stenosis (ICAS). A logistic regression model was used in the study to explore the relationship between computed tomography perfusion (CTP) parameters and delayed cerebral ischemia (DCI). In order to evaluate the improvement of model performance, prognostic-related perfusion parameters were added. This study found that assessing cerebral perfusion quantitatively may help diagnose patients suffering from both aSAH and ICAS. There is a lot of valuable information in this study, and I think it is comprehensive. I find the subject fascinating. To improve the scientific level of the article, the following significant corrections seem necessary.
1- The Abstract section should be formatted differently, and the numbers and subsection names should be removed.
2- The motivation behind the work should be clearly stated by the authors. The problem is not properly explained. The authors should demonstrate a scientific interest in the study's objectives and results.
3 - Compare the findings of this paper with those of similar papers. No clear indication of scientific merit or novelty can be found in the article.
4- Provide a table for the results in figure 2.There should be more information in the captions of the figures.
5- Summarize the main ideas learned and why they are valuable and exciting in a better conclusion.
6- A literature review of the benefits and drawbacks of existing methods and how the proposed method differs from them should be included in the "Introduction" section. It is also critical to clarify their motivations and contributions.
Author Response
Reviewer #2:
Comment 1:
The Abstract section should be formatted differently, and the numbers and subsection names should be removed.
Response 1:
Thanks for your comment.We revised in accordance with your comments.
Comment 2:
The motivation behind the work should be clearly stated by the authors. The problem is not properly explained. The authors should demonstrate a scientific interest in the study's objectives and results.
Response 2:
Thanks for your comment.In our clinical work, we see many aSAH patients coexistent with intracranial atherosclerotic stenosis, and for these patients, whether surgery would be beneficial and whether the concomitant lesions would cause ischemic events and have an unfavorable outcomes are the key events of interest in this study.These are reflected in line 43-50 of the manuscript.
Comment 3:
Compare the findings of this paper with those of similar papers. No clear indication of scientific merit or novelty can be found in the article.
Response 3:
Thanks for your comment.Previous studies have been more on carotid artery stenosis coexistent with unruptured aneurysm but less on ruptured aneurysm coexistent with intracranial atherosclerotic stenosis.The literature on ruptured aneurysm coexistent with intracranial atherosclerotic stenosis is also rarely.
Comment 4:
Provide a table for the results in figure 2.There should be more information in the captions of the figures.
Response 4:
Thanks for your comment. The subgroup analysis indicated that the female, old, and those underwent endovascular treatment were more likely to have poor outcome if their Tmax > 4.0s volume > 0 mL (Figure 2). There was no significant interaction between the perfusion parameter and these subgroups. The effect sizes of the volume in the moderate and severe intracranial atherosclerotic stenosis subgroups were unable to be calculated as all the patients with unfavorable outcomes were in the Tmax > 4.0s volume > 0 mL group.
Comment 5:
Summarize the main ideas learned and why they are valuable and exciting in a better conclusion.
Response 5:
Thanks for your comment. In practice, we do not know exactly how many patients with aneurysmal subarachnoid hemorrhage have combined intracranial artery stenosis, but our data give a prevalence of 25.9%, although it is only a single-center study.Although the incidence is not very high, we should focus on patients with aneurysmal subarachnoid hemorrhage coexistent intracranial atherosclerotic stenosis, and how to predict the prognostic outcome of these patients is the point of our study. The use of CTP-based RAPID software to assess the poor prognosis of these patients is the innovative point of our study.
Comment 6:
A literature review of the benefits and drawbacks of existing methods and how the proposed method differs from them should be included in the "Introduction" section. It is also critical to clarify their motivations and contributions.
Response 6:
Although Li compared their TAPS model with five other prediction models. TAPS is an easy-to-perform model that contains easily accessible indicators. In the derivation cohort, TAPS (AUC = 0.816) shows better discrimination than SAHIT (AUC = 0.802), FRESH (AUC = 0.784), Lai et al. 2020 (AUC = 0.742), HAIR (AUC = 0.727), and Mao et al. 2016 (AUC = 0.626). In the validation cohort, TAPS(AUC = 0.810) shows better discrimination than FRESH (AUC=0.799), HAIR (AUC = 0.796), Lai et al. 2020 (AUC = 0.749), and Mao et al. 2016 (AUC = 0.694). In the prospective cohort, TAPS(AUC = 0.803) shows better discrimination than FRESH (AUC = 0.796), Mao et al. 2016 (AUC= 0.606), and Lai et al. 2020 (AUC=0.546) .The SAHIT prediction models were based on patient data from different regions and settings of care and included 10 936 patients in the development set. They reliably estimated outcome in 3355 patients in the validation set.Notably, SAHIT showed better discrimination than TAPS in our validation and prospective cohorts. We acknowledge that SAHIT is an excellent prognostic model.
However, each model has its own advantages and disadvantages, and we thought that if we focused on comparing the advantages and disadvantages of various models in the introduction, it might increase the length of the article, so we did not describe this part.Finally, this was why we chose the SAHIT and TAPS models as the reference models in our paper.

Reviewer 3 Report
The paper reports about the employment of logistic regression models to investigate the association between computed tomography perfusion (CTP) parameters and 3-month neurological outcomes and delayed cerebral ischemia (DCI). The paper is interesting and well written. However, in my opinion, some concerns need to be addressed before publication:
1) Please add some information regarding the models’ development. Particularly, it should be clearly stated:
· the numerosity of each class identified.
· whether the classes were balanced (and in case they were not balanced whether some technique to balance the classes were used).
· whether an iteration procedure was implemented in order to reduce the risk of over fitting due to the choice of the folds.
· the number of input features (they should be less numerous with respect to the samples of each class).
2) Please, in the Results section, please report also the sensitivity and specificity of each model. Maybe, presenting the associated confusion matrix could be useful for the readers to have these information about the performance of the models.
3) In the discussion section the Authors should provide some possible future improvements (or comparison, if some similar models are already available) of the method relying on other neuroimaging techniques able to measure the cerebral perfusion (e.g., fMRI and fNIRS)
Author Response
Reviewer #3:
We are grateful for your positive comments, and we have now tried to address each of the cogent comments and suggestions with revision are as follows:
Comment 1:
Please add some information regarding the models’ development. Particularly, it should be clearly stated:
(1) the numerosity of each class identified.
(2) whether the classes were balanced (and in case they were not balanced whether some technique to balance the classes were used).
(3) whether an iteration procedure was implemented in order to reduce the risk of over fitting due to the choice of the folds.
(4) the number of input features (they should be less numerous with respect to the samples of each class).
Response 1:
(1) Thanks for your comment. The study population was described in the first line of Table 2.
(2) Thanks for your comment. The proportion of mRS>2 in our study was 21.9%, whose balance was acceptable in clinical studies. Therefore, no further methods were used to address the sample balance issue.
(3) Thank you for your comment and kindly reminder. We have indeed implemented an iteration procedure with repeated 100 times 10-fold cross-validation to reduce the risk of overfitting due to the choice of folds (using ‘caret’ package, version 6.0-93 in R). We have added a description of this procedure in the methods section of our revised manuscript.We have added“These models were trained using 10-fold cross-validation with the procedure iterated 100 times to reduce the risk of overfitting due to the choice of folds.”in Line 175-176.
(4) According to the EPV principle, at most 8 adjusted covariates could be appropriate. In this study, we adjusted for 10 statistically different variables in multivariable analysis 1 and additional 5 clinically important variables in multivariable analysis 2, which did not satisfy the EPV principle. Therefore, the results may not be robust enough. However, as the results are interpretable, these results were remained in the manuscript. And we have emphasized this issue in the limitation, we stated and added that “Fifth, the results should be interpret with caution as the conclusions were drawn after adjusting for numerous potential confounders, while the outcome events of this study did not meet the event per variable principle. The reliability and robustness of the results required further validation with larger sample size.” in Line 370-374.
Comment 2:
Please, in the Results section, please report also the sensitivity and specificity of each model. Maybe, presenting the associated confusion matrix could be useful for the readers to have these information about the performance of the models.
Response 2:
Thanks for your valuable comment. We have added the sensitivity, specificity, accuracy, PPV, NPV of these models as a supplementary table (Table S2), and have mentioned these information in the Results section (subsection 3.3).
“The cross-validated AUC did not differ significantly after the addition of the Tmax > 4.0s volume > 0 mL (SAHIT: P = 0.591; TAPS: P = 0.379). However, the cNRI and IDI showed that the perfusion predictor significantly improved the performance of the two models (P < 0.001 for all comparisons, Table 4). The evaluation metrics (sensitivity, specificity, accuracy, positive predictive value, and negative predictive value) derived from the confusion matrix of the four models were provided in Table S2.”in Line 230-232.
Table S2. The Sensitivity, Specificity, Accuracy, Positive Predictive Value (PPV), and Negative Predictive Value (NPV) of different models.
|
Model |
Sensitivity |
Specificity |
Accuracy |
PPV |
NPV |
|
SAHIT |
0.913 |
0.381 |
0.797 |
0.840 |
0.5512 |
|
SAHIT+CTP |
0.940 |
0.452 |
0.833 |
0.860 |
0.679 |
|
TAPS |
0.907 |
0.429 |
0.802 |
0.850 |
0.563 |
|
TAPS+CTP |
0.933 |
0.429 |
0.823 |
0.854 |
0.643 |
Abbreviations: CTP, Computed Tomography Perfusion; SAHIT, Subarachnoid Hemorrhage International Trialists; TAPS, Tiantan Aneurysmal Subarachnoid Hemorrhage Prognostic Scoring System.
Comment 3:
In the discussion section the Authors should provide some possible future improvements (or comparison, if some similar models are already available) of the method relying on other neuroimaging techniques able to measure the cerebral perfusion (e.g., fMRI and fNIRS)
Response 3:
Thanks for your comment. We have added “Near-infrared spectroscopy (NIRS) is a noninvasive technique that can provide a surrogate marker of changes in CBF and used to assess cerebral autoregulation in adults after subarachnoid hemorrhage. It seems likely that fNIRI and CTP will become a clinical tool in the foreseeable future, which will enable diagnosis and assess in an-eurysmal subarachnoid hemorrhage .” in Line 343-347.

Round 2
Reviewer 2 Report
It seems that the work has been improved as a result of the author's corrections.
Author Response
Thank you for your comments and suggestions.
Reviewer 3 Report
I appreciate the Authors responding to my comments. Yet, I still have reservations regarding the classes' balance. Specifically, the class mRS score <= 2 appears to be three times greater than the class mRS score >2. I advise to the Authors that they construct an iteration technique in which, at each iteration, a number of random samples of the most numerous class equal to the size of the smaller class are selected. So, all of the samples from the class with the greatest number of members are utilized in the classification phase, and the classes are balanced. This technique can lessen the potential bias a model may have towards the largest class. In reality, from the stated sensitivity and specificity values, it is plain that the model's good accuracy depends on its high sensitivity, but it is also evident that the model cannot accurately categorize the class with the fewest instances (the specificity is low). Due to the unbalanced distribution of the classes, the model has a bias against one of the classes. The proposed approach can solve this problem. Please report the results of this type of analysis.
Author Response
Reviewer #3:
Commen:I appreciate the Authors responding to my comments. Yet, I still have reservations regarding the classes' balance. Specifically, the class mRS score <= 2 appears to be three times greater than the class mRS score >2. I advise to the Authors that they construct an iteration technique in which, at each iteration, a number of random samples of the most numerous class equal to the size of the smaller class are selected. So, all of the samples from the class with the greatest number of members are utilized in the classification phase, and the classes are balanced. This technique can lessen the potential bias a model may have towards the largest class. In reality, from the stated sensitivity and specificity values, it is plain that the model's good accuracy depends on its high sensitivity, but it is also evident that the model cannot accurately categorize the class with the fewest instances (the specificity is low). Due to the unbalanced distribution of the classes, the model has a bias against one of the classes. The proposed approach can solve this problem. Please report the results of this type of analysis.
Response:
Thank you for your comments and suggestions. We appreciate your concerns about class balance and agree that it is an important issue to address. We have implemented the random under-sampling to balance our classes.
The description of this procedure was stated in the “2.6. Statistical Analysis” subsection: “These models were trained using 10-fold cross-validation with the procedure iterated 100 times to reduce the risk of overfitting due to the choice of folds. To address the issue of imbalanced outcome distribution, the random under-sampling procedure was implemented. This technique involved randomly selecting a subset of samples from the majority class to match the size of the minority class, which can lessen the potential bias of models toward the majority class.”
The corresponding results were reported in the “3.3. Prespecified subgroup analysis and additive value assessment” subsection, and provided as Table 4: “The cross-validated AUC did not differ significantly after adding the Tmax > 4.0s volume > 0 mL (SAHIT: P = 0.591; TAPS: P = 0.379). The evaluation metrics (sensitivity, specificity, accuracy, positive predictive value, and negative predictive value) derived from the confusion matrix of the four models were provided in Table S2. After the random under-sampling procedure, the overall sensitivity and specificity of models were balanced, and still, no significant addictive value was observed in these metrics (Table S3). However, the cNRI and IDI showed that the perfusion predictor significantly improved the performance of the two models (P < 0.001 for all comparisons, Table 4).”
Table S3. The Performance of different models after the random under-sampling procedure.
|
Model |
AUC (95%CI) |
Sensitivity |
Specificity |
Accuracy |
PPV |
NPV |
|
SAHIT |
0.777 (0.702-0.852) |
0.720 |
0.762 |
0.729 |
0.915 |
0.432 |
|
SAHIT+CTP |
0.779 (0.699-0.860) |
0.740 |
0.667 |
0.724 |
0.888 |
0.418 |
|
TAPS |
0.700 (0.608-0.791) |
0.747 |
0.643 |
0.724 |
0.882 |
0.415 |
|
TAPS+CTP |
0.701 (0.604-0.798) |
0.693 |
0.738 |
0.703 |
0.904 |
0.403 |
Abbreviations: AUC, area under the curve; CTP, computed tomography perfusion; NPV, negative predictive value; PPV, positive predictive value; SAHIT, Subarachnoid Hemorrhage International Trialists; TAPS, Tiantan Aneurysmal Subarachnoid Hemorrhage Prognostic Scoring System.
